# Skeletal Anterior Open Bite Attenuates the Chewing-Related Increase in Brain Blood Flow

**DOI:** 10.3390/dj12060161

**Published:** 2024-05-27

**Authors:** Chihiro Kariya, Hiroyuki Kanzaki, Masao Kumazawa, Saaya Sahara, Kana Yoshida, Yuri Inagawa, Fumitaka Kobayashi, Hiroshi Tomonari

**Affiliations:** Department of Orthodontics, School of Dental Medicine, Tsurumi University, Yokohama 230-8501, Japan; chihirortho.k@gmail.com (C.K.); qqmw7fm9k@abelia.ocn.ne.jp (M.K.); snowdropsya@gmail.com (S.S.); yoshidakana0x0@gmail.com (K.Y.); yourina0314@gmail.com (Y.I.); fkobayashi.tsu@gmail.com (F.K.); tomonari-h@tsurumi-u.ac.jp (H.T.)

**Keywords:** brain blood flow, open bite, jaw deformity, masticatory function, brain function, near-infrared spectroscopy (NIRS), surgical orthodontic treatment, chewing, mastication

## Abstract

The masticatory function of patients with skeletal anterior open bite (OPEN) is reported to be impaired compared with that of patients with normal occlusion (NORM). In this study, we compared brain blood flow (BBF) in patients with OPEN and NORM and investigated the factors related to BBF during mastication in patients with OPEN. The study included 17 individuals with NORM and 33 patients with OPEN. The following data were collected: number of occlusal contacts, jaw movement variables during mastication, and BBF measured with functional near-infrared spectroscopy during chewing. The number of occlusal contacts, maximum closing and opening speeds, closing angle, and vertical amplitude were smaller in the OPEN than in the NORM group. Interestingly, BBF increased less in the OPEN group. Correlation analysis revealed that several parameters, including number of occlusal contacts and closing angle, were correlated with changes in BBF during mastication. These results suggest that not only occlusion but also jaw movement variables and factors related to masticatory muscles contribute to the chewing-related increase in BBF. In conclusion, BBF increases less during mastication in patients with OPEN than in those with NORM. In addition, the higher increase in BBF is correlated with jaw movement. Together, we discovered that OPEN exhibits significant adverse effects not only on masticatory function but also on brain function.

## 1. Introduction

Anterior open-bite malocclusion, which is defined as the absence of vertical overlap of the incisors when the teeth of the buccal segment are in occlusion, is one of the most difficult malocclusions to treat because of the high frequency of relapse [1,2,3]. Anterior open bite has a multifactorial etiology, including skeletal, dental, respiratory, neurologic, and habitual components [1]. In addition, some syndromes, such as Treacher–Collins Syndrome [4], Apert syndrome or Crouzon syndrome [5], and Robin sequence [6], exhibit the open-bite phenotype. Systemic diseases, including Duchenne muscular dystrophy, also relate to open-bite malocclusion [7]. Interestingly, drug-induced open bite is also reported [8].

Patients with a skeletal anterior open bite (OPEN) may have some or all of the following cephalometric features [9]: high-angle skeletal pattern with increased Frankfort mandibular plane angle [10], pronounced ante-gonial notching, recessive chin, reduced inter-incisal angle, reduced inter-molar angle, and increased lower anterior facial height [11]. OPEN can lead to a number of problems, such as difficulty in biting and chewing [12], speech problems [13], and even jaw pain and temporomandibular joint (TMJ) disorders [14]. It can also have negative aesthetic implications, such as a gummy smile [15]. Patients with OPEN have relatively weak muscle activity in masticatory muscles [16] and have difficulty closing their lips [17]. Thus, OPEN causes not only aesthetic issues but also numerous functional problems.

In a previous study, we compared the characteristics of patients with mandibular prognathism, a common jaw deformity in Japan, and those with normal occlusion (NORM) and found that mandibular prognathism attenuates the chewing-related increase in brain blood flow (BBF) [18,19]. Otsuka et al. also studied the relationship between malocclusion and BBF and found that the forced malocclusion model augmented participants’ stress level and that prefrontal cortex activation was correlated with the visual analog scale score for discomfort [20]. Artificial occlusal disharmony is also reported to affect BBF [21,22]. In addition to these clinical studies, basic research using animal models clearly demonstrated that reduced masticatory stimulation during growth periods induced a decline in brain functions such as memory and learning [23]. Furthermore, the experimental unilateral anterior crossbite in experimental animals revealed a negative impact on the motor nuclei in the brain stem via the trigeminal mesencephalic nucleus [24]. Occlusal disharmony induced the accumulation of molecules such as amyloid-β and phosphorylated tau, which resulted in protection against transient dementia in young but not older individuals [25]. Taken together, these findings show that malocclusion influences brain function and BBF.

To date, no study has examined the relationship between OPEN and chewing-related changes in BBF. Therefore, in the present study we hypothesized that OPEN might influence BBF during chewing and performed a clinical study to clarify this hypothesis. We compared the characteristics of patients with OPEN and those of patients with NORM, especially from the perspective of BBF, and aimed to elucidate the factors responsible for the attenuated increase in BBF during chewing in OPEN.

## 2. Materials and Methods

### 2.1. Ethical Approval

This cross-sectional study was approved by the ethics committee of Tsurumi University, School of Dental Medicine (approval numbers: 1316 and 121005), and conformed to the principles of the Declaration of Helsinki. Written informed consent was obtained from all participants before study commencement. This was a human observational study and conformed with the STROBE guidelines.

### 2.2. Sample Size Calculation

Prior to the experiment, we estimated the required sample size using the G Power 3.1.9.4 program (Universitat Kiel, Kiel, Germany), with the following parameter settings: effect size = 0.8, alpha (significance level) = 0.05, power = 0.8, and allocation ratio of the groups = 2 (1:2 ratio for NORM–OPEN). The minimum total sample size was computed as 46, and our total sample number (50) was above the estimated number.

### 2.3. Participants

The present study included participants with NORM and OPEN. The NORM group (*n* = 17) comprised 4 men and 13 women with a median age of 19.0 years (interquartile range [IQR], 19.0–20.0 years).

The inclusion criteria were as follows:No defects other than missing third molars;Adequate overjet and overbite (2–4 mm each, proper occlusion of anterior teeth);Skeletal and dental midline deviation of less than 1.0 mm;No functional symptoms, such as TMJ disorder;No history of orthodontic treatment;Angle Class I molar relationship (normal molar relationship in anteroposteriorly);Normal intermaxillary relationship in cephalometric x-ray analysis (A point-nasion-B point [ANB] angle, 3.1 ± 2.5°; mandibular plane to Frankfort horizontal [FH] angle: 26.6 ± 6.7°).

Participants in the NORM group were recruited from volunteers. As to the status of TMJ, the Diagnostic Criteria for Temporomandibular Disorders (DC/TMD) were used to eliminate the subjects with TMJ problems based on DC/TMD in both NORM and OPEN.

The OPEN group (*n* = 33) comprised 9 male and 24 female patients with a median age of 22.0 years (IQR, 18.0–32.0 years) being treated at the Tsurumi University Hospital. Patients met the following inclusion criteria:Skeletal open bite (32° or higher in the mandibular plane to FH angle) requiring orthognathic treatment;No congenital abnormality;No missing teeth other than third molars;Overbite, which indicates vertical overlap between upper and lower anterior teeth, less than 0 mm;No TMJ abnormality;No previous orthodontic treatment.

The subjects we selected consisted mainly of patients with open bite in anterior and premolar regions. In addition, some cases exhibited anterior open bite, from canine to canine.

The classifications of the anteroposterior skeletal relationship according to the Japanese standard ANB value [26] were as follows: Class I (normal intermaxillary relationship), six patients; Class II (maxillary protrusion and mandibular retrognathism), nine patients; and Class III (maxillary retrognathism and mandibular protrusion), 18 patients.

### 2.4. Examination of Occlusal Contacts with Silicone Materials

The examination of occlusal contacts is described elsewhere [18]. Briefly, occlusal contacts were recorded with vinyl polysiloxane impression material (PerfectIM systems; J Morita, Tokyo, Japan) and scanned, and the number of occlusal contacts was counted with ImageJ software 1.53u (Bethesda, MD, USA).

### 2.5. Recording of Jaw Movement

Jaw movement during mastication was recorded with the Gnatho-hexagraph III (GC Co., Ltd., Tokyo, Japan). For the recording, participants were fitted with a head frame and face bow, the mandibular condyle, porion, orbitale, ANS, mandibular first molars, and mandibular incisors were pointed, and the Frankfurt plane was also set. Participants were then asked to chew gum, and the rhythm and path of jaw movement, including preferential chewing and left- and right-side chewing, were measured and recorded in the x–y, y–z, and z–x planes. As to preferential chewing, the participant chewed freely on the person’s preferred side of the molars [27]. In each chewing task, the mastication cycles before stabilization of jaw movement were omitted from the analysis, and only the 15 to 20 cycles after stabilization were included. The analyzed parameters are described elsewhere [27]. Briefly, in the opening cycle, the jaw movement was analyzed from 0.7 mm below the most superior position to the most inferior position of the masticatory cycle, and in the closing cycle, from the most inferior point of the masticatory cycle to 0.7 mm below the most superior position. The maximum velocities of the opening and closing phases for each chewing cycle were calculated according to the conventional method. The closing and opening angles (degree) and cycle width (mm) were determined at the vertical slice level 2.0 mm below the maximum intercuspal position in the frontal view.

### 2.6. Measurement of BBF

Measurement of BBF is described elsewhere [13]. Briefly, BBF was measured with functional near-infrared spectroscopy (fNIRS) (OEG-16 device; Spectratech, Tokyo, Japan), which measures oxidized hemoglobin in the prefrontal cortex with 16 channels. The center of the probe matrix was placed on Fpz (midpoint between Fp1 and Fp2) in accordance with the international 10/20 system used in electroencephalography. The probe in the bottom left corner was placed around F7, and the right probe was placed around F8.

In the fNIRS measurement, a simple block design consisting of one control and experimental task conditions, with adequate resting intervals over 30 s, was used. The control task was a calculation (subtraction) task, in which the subjects were asked to verbally reply with the serial subtraction of 7 from 100. The experimental tasks constituted chewing of CRT paraffin wax (Ivoclar Vivadent, Tokyo, Japan) on their preferential side. The fNIRS data were analyzed using oxidized hemoglobin values to infer changes in BBF in the 16 channels. The maximum value of the oxy-Hb in each channel during each task was used for the analysis. The value of oxy-Hb during the calculation task was used as a positive control, and the ratio of the value of oxy-Hb in each channel during the chewing tasks to the mean value of oxy-Hb in the calculation task was used in the comparison.

We focused on channels 1 to 4 and 13 to 16, which contained the inferior frontal gyrus, because the inferior frontal gyrus has been reported to play an important role in cognitive function [28,29].

### 2.7. Statistical Analysis

Statistical analyses were performed with SPSS Statistics (IBM, Tokyo, Japan). First, data were evaluated for normal distribution with D’Agostino’s chi-square test and the Kolmogorov–Smirnov test. For parametric data, Student’s *t* test was used to test for statistical significance, and for nonparametric data, the Mann–Whitney U test was used. The Fisher exact test was used to assess whether the ratio of men to women was different between the groups. Parametric data were expressed as mean ± standard deviation, and nonparametric data as median and quartiles; a *p* value of less than 0.05 was considered to indicate a statistically significant difference. Spearman’s rank correlation coefficient was used to examine the correlation of each factor with the chewing-related increase in BBF.

## 3. Results

### 3.1. Characteristics of NORM and OPEN Groups

Table 1 shows the characteristics of the NORM and OPEN groups. There was no difference in the ratio of men to women between the groups (Table 2). However, cephalometric analysis revealed differences between the groups, such as in the mandibular plane to FH. Furthermore, the OPEN group had fewer occlusal contacts than the NORM group. The mean age in the OPEN group was significantly higher than in the NORM group (Table 3).

### 3.2. OPEN Had a Negative Impact on Masticatory Jaw Movement

Compared with the NORM group, the OPEN group showed some differences in parameters of masticatory jaw movement, i.e., the maximum closing and opening speeds, the closing angle, and the vertical amplitude were significantly smaller (Table 4). However, the cycle duration was similar in both groups. These results suggest that in the OPEN group, jaw movement was slow, with a chopping stroke.

### 3.3. OPEN Had a Negative Impact on the Chewing-Related Increase in BBF

The OPEN group showed a smaller chewing-related increase in BBF than the NORM group (Figure 1). There was no statistically significant difference between the sides in each group. In addition, this reduction in OPEN was observed on both the right and left sides of the inferior frontal gyrus.

Furthermore, the value in the OPEN group was smaller than that in patients with mandibular prognathism in our previous study (0.131 vs. 0.164, respectively) [30]. These results suggest that not only mandibular prognathism but also OPEN attenuates the chewing-related increase in BBF. Interestingly, the reduction in the chewing-related increase in BBF was greater in OPEN than in mandibular prognathism.

### 3.4. Factors Associated with the Chewing-Related Increase in BBF

The results of the Spearman’s rank correlation coefficient analysis to clarify the factors associated with the chewing-related increase in BBF are shown in Table 5. Because age was significantly different between the two groups, we first analyzed whether age was associated with BBF; we found no correlation, indicating that the difference in BBF between groups was not due to the difference in age.

Among the 28 factors, some were correlated with a change in BBF. In the right inferior frontal gyrus, maximum closing velocity (r = −0.409), maximum opening velocity (r = −0.383), and closing angle (r = −0.386) during right-side mastication and maximum closing velocity (r = −0.479) and maximum opening velocity (r = −0.449) during left-side mastication were inversely correlated with BFF; furthermore, maximum opening during preferential chewing (r = 0.350) was positively correlated with BF.

In the left inferior frontal gyrus, negative correlations were found for the number of occlusal contacts (r = −0.346) and closing angle during right-side chewing (r = −0.537), and a positive correlation was found for the closing angle during left-side chewing (r = 0.394). The Frankfort mandibular plane angle (FMA) and overbite were not correlated with the change in BBF.

These results suggest that the difference in jaw movement during mastication influenced the chewing-related change in BBF in the OPEN group.

## 4. Discussion

In this study, we found that the chewing-related increase in BBF was smaller in patients with OPEN than in those with NORM. In addition, the change in BBF during chewing was correlated with certain factors, such as the number of occlusal contacts and jaw movement. Our results suggest that OPEN has a significant negative impact on jaw movement variables and factors related to mastication, which results in a smaller increase in BBF during chewing. Therefore, these findings suggest that orthodontic treatment for skeletal open bite patients not only has clinical significance in improving jaw and oral functions but may also have the potential to alleviate the adverse effects on brain function.

Because the present study used fNIRS to measure the change in BBF, we could only observe changes in the prefrontal cortex. Narita et al. showed that chewing gum increases BBF not only in the prefrontal cortex but also in areas of the posterior cortex [28], so it is possible that different reactions may occur in other areas. Previously, we revealed that jaw deformity has a negative impact on BBF during chewing, especially in the inferior frontal gyrus [18,19], an area reported to play an important role in cognitive function [29,31]. Therefore, we wanted to explore the effect of OPEN primarily in the inferior frontal gyrus. Further exploration with functional magnetic resonance imaging could clarify the changes in other areas of the brain.

The chewing-related increases in BBF observed in the inferior frontal gyrus were lower in the OPEN than in the NORM group. Previously, we compared patients with mandibular prognathism and individuals with NORM and found that the patients had a smaller chewing-related increase in BBF in the inferior frontal gyrus [18,19]. We found no statistically significant difference in the ratio of men to women and the median age between the mandibular prognathism patients in that study and the OPEN patients in the present study (data not shown), so we compared the two groups and found that the reduction in the chewing-related increase in BBF had a tendency to be greater in the OPEN patients than in the mandibular prognathism patients in the previous study [30]. These results suggest that OPEN may have a greater negative impact on BBF than mandibular prognathism. Coincidentally, Restrepo et al. also reported the negative impact of anterior open bite on brain cortex activity [32]. Further detailed comparisons of the effects of different types of jaw deformity on BBF and potentially on brainn function are required.

Our data clearly demonstrated a smaller chewing-related increase in BBF in the inferior frontal gyrus in patients with OPEN, suggesting that this jaw deformity may have negative effects on inferior frontal gyrus function. Because the inferior frontal gyrus has been reported to play an important role in cognitive function [29,31], OPEN may negatively influence cognitive function. Coincidentally, animal experiments clearly demonstrated that reduced masticatory stimulation during growth periods induced a decline in brain functions such as memory and learning [23], which supports our hypothesis that malocclusion-related retardation in brain blood flow during mastication influences brain function. To clarify the issue, a study is currently underway to determine whether and how jaw deformity affects cognitive function.

In OPEN patients, masticatory jaw movement was characterized by a chopping stroke, in other words, a small lateral movement during mastication and less vertical movement. The finding of a chopping stroke was consistent with previous reports [33,34,35]. A chopping stroke is also observed in the scissor-bite side of patients with posterior scissor bite and patients with mandibular prognathism [36]. One possible explanation why chopping stroke is observed in these patients may be the interference of the occlusal surface between the molars [27]. Another explanation of chopping stroke in OPEN patients may be the lack of lip seal. Shima et al. reported that masticatory motion with open lips is associated with chopping strokes, and they suggested that abnormal function of the lips, jaw, masticatory muscles, and tongue interferes with masticatory efficiency [33].

Statistical correlation analysis revealed that the maximum opening velocity, closing angle, and closing velocity were negatively correlated with BBF. The inverse correlation between maximum opening velocity and blood flow in the prefrontal cortex during chewing suggests that the increase in cerebral blood flow may be attenuated by impaired mouth opening. While temporomandibular joint (TMJ) disorders are a common cause of restricted jaw movement [36], our study population did not include patients with TMJ issues, indicating that other factors may contribute to this finding. Unexpectedly, we also observed an inverse correlation between closing velocity and blood flow, which contradicts a previous study reporting that gum chewing induced a greater increase in cerebral blood flow compared to a tapping task [37]. We had hypothesized that patients with an adequate grinding pattern during mastication would exhibit higher cerebral blood flow during chewing. Furthermore, another study demonstrated that the regional increase in brain neural activity during gum chewing was associated with the hardness of the gum, suggesting that greater masticatory jaw movement leads to higher cerebral blood flow [30]. Despite these contradictory results, our findings revealed that a smaller closing angle, indicative of a chopping stroke pattern, was associated with a greater increase in cerebral blood flow. One possible explanation for this phenomenon is that patients had already optimized their jaw movements over many years, even if these movements were suboptimal, at the time of the initial examination. Comparing pre- and post-treatment data would help elucidate this issue. As surgical orthodontic treatment for patients with skeletal open bite improves masticatory jaw function [38] and may also enhance the chewing-related increase in cerebral blood flow. We are currently collecting post-treatment data and intend to publish these results in the future to further investigate the impact of surgical orthodontic treatment on the relationship between masticatory function and cerebral blood flow.

The study has some limitations. Regarding the intermaxillary relationship in the OPEN group, we could not limit the subjects to Class I patients because of the small number of participants. As to this sample size issue, it seems that the number of the subjects is small, though our power analysis revealed that the present number of the subjects was above the estimated number. Therefore, we presumed that the number of the subjects in this study was sufficient. However, correlation analysis gave quite few statistically significant results, which indicates that there is high chance of obtaining more significant correlations with an increase in subjects. Further research using a higher number of subjects is necessary. In addition, we had no data on the comparison of or relationship with masticatory muscle activity in this study. Research clearly indicates low masticatory muscle activity in open bite patients as compared to subjects with normal occlusion [31,39], which leads to the idea that the brain blood flow is augmented by masticatory muscle activity. Indeed, the relationship between masticatory muscle activity and brain blood flow have been reported [37,40,41]. Therefore, exploration to clarify the relationship between masticatory muscle activity and brain blood flow in open bite subjects is necessary in the future.

This study focused on the vertical intermaxillary relationship, and participants were selected primarily by the results of cephalometric analysis, the mandibular plane to FH angle, i.e., FMA, and the results of dental study cast analysis, including overbite values. Therefore, the OPEN group consisted of a mixture of various antero-posterior intermaxillary relationships. In the OPEN group, almost half of the patients exhibited mandibular prognathism, which might explain why the results were similar to our previous study [18,19]. Therefore, a future study needs to examine the extent to which the anteroposterior skeletal relationship influences the chewing-related increase in BBF. In addition, future studies should aim to address the limitations of the present work and expand upon these initial findings. First, to clarify the influence of the antero-posterior skeletal relationship on chewing-related increases in BBF, studies with larger sample sizes that allow for stratification by skeletal classification are needed. Additionally, longitudinal studies assessing patients before and after surgical orthodontic treatment would provide valuable insights into how correction of skeletal open bite impacts jaw function, masticatory patterns, and BBF during mastication. Finally, the use of whole-brain imaging techniques such as functional magnetic resonance imaging could elucidate potential changes in brain regions beyond the prefrontal cortex. By building upon the current findings through these avenues of research, we can gain a more comprehensive understanding of the complex interplay between craniofacial morphology, masticatory function, and brain activity.

## 5. Conclusions

We discovered that patients with OPEN have a smaller chewing-related increase in BBF than individuals with NORM. In addition, the extent of the increase in BBF during chewing is correlated with the number of occlusal contacts and jaw movement. Our results suggest that OPEN has a significant negative impact on oral functions, including jaw movement, which results in a smaller increase in BBF during chewing.

## Figures and Tables

**Figure 1 dentistry-12-00161-f001:**
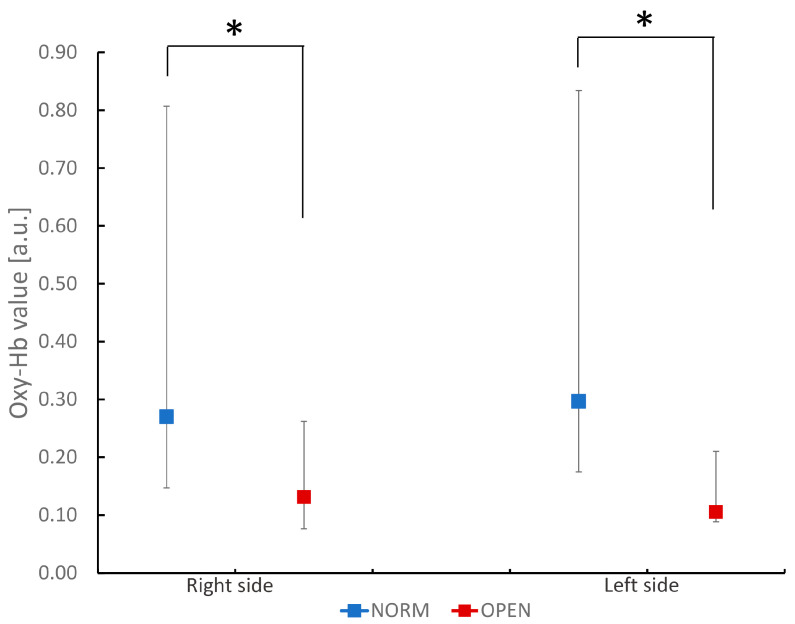
Comparison of oxidized hemoglobin (Oxy-Hb) values in the bilateral inferior frontal gyrus during chewing task in participants with normal occlusion and those with a skeletal anterior open bite. The figure shows oxidized hemoglobin values during the chewing task. Values are ex-pressed as median and quartiles of channels 1 to 4 (right side) and 13 to 16 (left side) in each group. * *p* < 0.05 between groups (Mann–Whitney U test).

**Table 1 dentistry-12-00161-t001:** Comparison of characteristics in participants with a skeletal anterior open bite and those with normal occlusion.

	OPEN (*n* = 33)		NORM (*n* = 17)		*p* Value
	Mean	S.D.	Mean	S.D.	
FMA, °	38.13	4.95	26.64	6.70	**
Overbite, mm	−3.08	0.53	2.92	0.69	**
Occlusal contacts, *n*	13.20	7.40	25.40	1.80	**

** *p* < 0.01. OPEN: skeletal anterior open bite; NORM: normal occlusion; S.D.: standard deviation.

**Table 2 dentistry-12-00161-t002:** Comparison of the ratio of men to women in participants with a skeletal anterior open bite and those with normal occlusion.

	Men	Women	Total
OPEN	9	24	33
NORM	4	13	17
Total	13	37	50

There was no statistically significant difference in the ratio of men to women between the groups (Fisher’s exact test).

**Table 3 dentistry-12-00161-t003:** Comparison of the median age of participants with a skeletal anterior open bite and those with normal occlusion.

	OPEN (*n* = 33)			NORM (*n* = 17)			*p* Value
	Median	25%	75%	Median	25%	75%	
Age, y	22.0	18.0	32.0	19.0	19.0	20.0	*

* *p* < 0.05.

**Table 4 dentistry-12-00161-t004:** Comparison of jaw movement variables during chewing of gum in participants with a skeletal anterior open bite and those with normal occlusion.

	OPEN (*n* = 33)		NORM (*n* = 18)		*p* Value
	Mean	S.D.	Mean	S.D.	
Right side chewing					
Cycle duration	0.8	0.2	0.8	0.2	-
Max. closing speed	74.7	22.6	99.8	22.8	**
Max. opening speed	68.7	21.8	99.1	21.6	**
Closing angle	19.7	18.6	37.5	9.7	**
Cycle width	2.2	1.4	2.6	1.1	-
Max. open	12.5	3.9	15.6	1.5	**
Left-side chewing					
Cycle duration	0.8	0.2	0.8	0.2	-
Max. closing speed	74.5	27.8	101.0	31.2	**
Max. opening speed	68.5	22.0	101.4	24.2	**
Closing angle	22.5	14.2	36.7	11.5	**
Cycle width	2.5	1.6	2.5	1.2	-
Max. open	12.6	4.1	15.2	2.3	**

** *p* < 0.01.

**Table 5 dentistry-12-00161-t005:** Correlations of various parameters with increase in brain blood flow in participants with a skeletal anterior open bite and those with normal occlusion.

Ch. 1–4 (Right Frontal Gyrus): Brain Blood Flow While Chewing Paraffin Wax				
	r		*p* value	
Age	0.05		NS	
FMA	−0.05		NS	
Overbite	−0.23		NS	
Number of occlusal contacts	0.03		NS	
	Right-side chewing		Left-side chewing	
	r	*p* value	r	*p* value
Rhythm CV	0.28	NS	0.18	NS
Cycle duration	0.08	NS	0.00	NS
Max. closing speed	−0.409	*	−0.479	**
Max. opening speed	−0.383	*	−0.449	**
Path CV	0.04	NS	0.32	NS
Closing angle	−0.386	*	0.04	NS
Cycle width	0.20	NS	0.10	NS
Max. open	−0.31	NS	−0.32	NS
Ch. 13–16 (left frontal gyrus): brain blood flow while chewing paraffin wax				
	r		*p* value	
Age	−0.26		NS	
FMA	−0.20		NS	
Overbite	−0.07		NS	
Number of occlusal contacts	−0.346		*	
	Right-side chewing		Left-side chewing	
	r	*p* value	r	*p* value
Rhythm CV	0.28	NS	0.20	NS
Cycle duration	−0.08	NS	−0.08	NS
Max. closing speed	−0.13	NS	−0.27	NS
Max. opening speed	−0.04	NS	−0.19	NS
Path CV	−0.22	NS	−0.06	NS
Closing angle	−0.537	**	0.394	*
Cycle width	0.28	NS	0.19	NS
Max. open	0.03	NS	−0.04	NS

* *p* = 0.05; ** *p* = 0.01. CV is coefficient variation, which is calculated as standard deviation divided by mean.

## Data Availability

The data that support the findings of this study are available from the corresponding author upon reasonable request.

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
