# Peer review of "Skeletal Anterior Open Bite Attenuates the Chewing-Related Increase in Brain Blood Flow"

_dentistry, 2024, doi:10.3390/dj12060161_

Round 1

Reviewer 1 Report

Comments and Suggestions for Authors

1.      Statistics are presented adequately. Therefore, sample size calculation (power analysis) is not described. Authors should add the specific tool used if exists.

2.      Authors at the conclusion section, describe the future research direction should be followed.

3.      The limitation of the current study is the small sample size.

Author Response

Thank you very much for fruitful discussion and suggestion. We revised our manuscript according to the reviewers’ comments, and detailed response are written below.

Comments and Suggestions for Authors

  1. Statistics are presented adequately. Therefore, sample size calculation (power analysis) is not described. Authors should add the specific tool used if exists.

---Response: Thank you very much for kind suggestion. As suggested, we added the description regarding sample size calculation in the materials and methods section, 2.2. Participants.

  1. Authors at the conclusion section, describe the future research direction should be followed.

---Response: Thank you very much for precious comment. As to the future research direction, we already written the sentence, “Therefore, a future study needs to examine the extent to which the anteroposterior skeletal relationship influences the chewing-related increase in BBF.”, in the last paragraph of the discussion. Generally, conclusion should describe the most important point of the manuscript, we dared to describe the suggested future research direction not in the conclusion but in the last paragraph of the discussion. In addition, we added the description about the issue in details.

  1. The limitation of the current study is the small sample size.

---Response: Thank you for criticizing the point. It seems that the number of the subject is small, though our power analysis revealed that the present number of the subject was above the estimated number. Therefore, we presumed that the number of the subject in this study is sufficient. However, correlation analysis gave quite few statistical significance, which indicates that there is high chance to obtain more significant correlation with increase of the subjects. We added the sentence regarding this sample size issue in the discussion.

Reviewer 2 Report

Comments and Suggestions for Authors

Thank you for this very interesting manuscript. I really encourage you to deepen the topic because it is very important.

I would like to give you some advices if you like:

can you add the power of the study( sample size)?

I would like to know if the kind of open bite is the same in all the patients? In particualr the amount? is it only anterior or lateral as well?

Is it only skeletal open bite, not dental one? because i can see that you report a high number of FMA.

Did you have any inclusion or exclusion criteria?

You affirm that these patients are negative for  TMJ problems, how was this assessed?

Another aspect, this difference that you found can not be tied to the fact that the normal group is with lower FMA and so with a lower vertical dimension and stronger facial muscles that increase the blood flow?

Thank you

Author Response

Thank you very much for fruitful discussion and suggestion. We revised our manuscript according to the reviewers’ comments, and detailed response are written below.

Comments and Suggestions for Authors

Thank you for this very interesting manuscript. I really encourage you to deepen the topic because it is very important.

---Response: I appreciate your supportive remarks.

I would like to give you some advices if you like:

can you add the power of the study( sample size)?

---Response: Thank you very much for kind suggestion. As suggested, we added the description regarding sample size calculation in the materials and methods section, 2.2. Participants.

I would like to know if the kind of open bite is the same in all the patients? In particualr the amount? is it only anterior or lateral as well?

---Response: Thank you for confirming the point. The subjects we selected were consisted mainly with the patients who has openbite at anterior and premolar region. In addition, some cases exhibit anterior openbite, from canine to canine. The overbite value measured at incisors were -3.08 ± 0.53 mm.

Is it only skeletal open bite, not dental one? because i can see that you report a high number of FMA.

---Response: Thank you very much for criticising the point. Because we wanted to select the subject with skeletal openbite, we included the subject with high FMA. There was statistically significant difference in FMA value between the control group (NORM; normal occlusion) and OPEN group (openbite group). Therefore, we believe that the subjects in OPEN group consisted with skeletal openbite cases.

Did you have any inclusion or exclusion criteria?

---Response: Thank you very much for fruitful critique. We have described inclusion criteria in the materials and methods section, 2.2. Participants. To augment readability, we modified the part. 

You affirm that these patients are negative for  TMJ problems, how was this assessed?

---Response: Thank you very much for kind confirmation. We routinely use Diagnostic Criteria for Temporomandibular Disorders (DC/TMD) to examine the TMJ problems. The subjects were negative for TMJ problem based on DC/TMD.

Another aspect, this difference that you found can not be tied to the fact that the normal group is with lower FMA and so with a lower vertical dimension and stronger facial muscles that increase the blood flow?

---Response: Thank you very much for precious suggestion. Unfortunately, we did not have the data of muscle activity. As many papers reported the weak masticatory muscle activity in the patient with openbite, which support the reviewer’s idea that brain blood flow would relate to the masticatory muscle activity. Further researches are necessary to elucidate the relationship between masticatory muscle activity and brain blood flow. This issue is discussed in the discussion.

Thank you

Reviewer 3 Report

Comments and Suggestions for Authors

The present study, which evaluates the chewing-related BBF in patients with AOB and compared it with normal occlusion patients, is interesting. It is quite well structured and written in a quite understandable English.

The study showed that BBF increases less during mastication in patients with AOB than in those with normal occlusion.  However, the BBF was evaluated only in the prefrontal cortex, and it is not demonstrated that the reduction in the BBF in the AOB patients is enough to affect brain function. Maybe you should comment about the second point in the discussion.

I have some other comments:

MATERIALS AND METHOD

I would describe a bit more the measurement of BBF, particularly about the channels that are then mentioned later in the text.

Did you perform a a-priori sample size calculation for the study?

RESULTS

The description of Figure 1 in the text is not clear (page 5 lines 159-161)

The description of Table 5 in the text (page 5 from 173-177) does not correspond with values in the table itself.

DISCUSSION

The discussion is a bit repetitive. And the paragraph at page 7 between 234 and 254 is not clear to me. You should try to reduce repetition making the discussion a bit shorter and re-write the paragraph I cited.

Comments on the Quality of English Language

Moderate editing of English language required

Author Response

Thank you very much for fruitful discussion and suggestion. We revised our manuscript according to the reviewers’ comments, and detailed response are written below.

Comments and Suggestions for Authors

The present study, which evaluates the chewing-related BBF in patients with AOB and compared it with normal occlusion patients, is interesting. It is quite well structured and written in a quite understandable English.

---Response:  I appreciate your supportive remarks.

The study showed that BBF increases less during mastication in patients with AOB than in those with normal occlusion.  However, the BBF was evaluated only in the prefrontal cortex, and it is not demonstrated that the reduction in the BBF in the AOB patients is enough to affect brain function. Maybe you should comment about the second point in the discussion.

---Response: Thank you very much for fruitful suggestion. At this time, there is no evidence that shows attenuated brain blood flow during chewing in openbite subjects induce retarded brain function. Actually, we are now exploring the issue with measuring the cognitive function using eye tracking-based cognitive assessment device (https://www.ai-brainscience.co.jp/technology-eyetracking-en/). We will clearly answer the issue with the data in the near future. The discussion part is rewritten regarding the required future experiments on brain function.

I have some other comments:

MATERIALS AND METHOD

I would describe a bit more the measurement of BBF, particularly about the channels that are then mentioned later in the text.

---Response: Thank you very much for kind suggestion. As suggested, we explained the channels we observed in the materials and methods.

Did you perform a a-priori sample size calculation for the study?

---Response: Thank you very much for confirming the issue. We performed power analysis and the details were described in the materials and methods section.

RESULTS

The description of Figure 1 in the text is not clear (page 5 lines 159-161)

---Response: Thank you very much for pointing out the issue. We write the figure legend of figure 1 as maintext after figure 1 by mistaken.

The description of Table 5 in the text (page 5 from 173-177) does not correspond with values in the table itself.

---Response: Thank you very much for pointing out the issue. We forgot to indicate the table containing the data by mistaken. We now put the missing data, following the previous table 5.

DISCUSSION

The discussion is a bit repetitive. And the paragraph at page 7 between 234 and 254 is not clear to me. You should try to reduce repetition making the discussion a bit shorter and re-write the paragraph I cited.

---Response: Thank you very much for criticising the issue. As criticized, we omit the redundant part in the discussion as much as possible. In addition, the unclear part for the reviewer is rewritten to increase the understandability.

Comments on the Quality of English Language

Moderate editing of English language required

---Response: Thank you very much for kind suggestion. We already used editing service for scientific English language. We sent the certificate for review process.

Reviewer 4 Report

Comments and Suggestions for Authors

Skeletal anterior open bite attenuates the chewing-related increase in brain blood flow 

The title of the article is appropriately selected and denotes the study performed.

The aim of the study is clear.

The authors reviewed well what is already written and investigated in the literature till 2021. 

The methodology is clear enough and the number of participating subjects is fair.

The results are well written, tables and graphs are clear and readable.

The discussion section is very well written and elaborated good with other researchers results.

The authors pointed out to the study limitation about inclusion of the anteroposterior skeletal relationship influences the chewing-related increase in BBF. 

I have two thoughts about the research in general:

1- In the introduction and discussion: why didn’t you have any correlation to those syndromes that OPEN is one of the signs and symptoms?!

2- In the conclusion: How do you know that OPEN has a significant negative impact on BBF?!, What about the possibility that the BBF is the Causative factor of OPEN?!

The references are relatively recent (2021) and relevant in addition to this are well written and arranged and highly related to the study. Is there no references related to the study in the last 3 years?!

Author Response

Thank you very much for fruitful discussion and suggestion. We revised our manuscript according to the reviewers’ comments, and detailed response are written below.

Comments and Suggestions for Authors

Skeletal anterior open bite attenuates the chewing-related increase in brain blood flow 

The title of the article is appropriately selected and denotes the study performed.

The aim of the study is clear.

The authors reviewed well what is already written and investigated in the literature till 2021. 

The methodology is clear enough and the number of participating subjects is fair.

The results are well written, tables and graphs are clear and readable.

The discussion section is very well written and elaborated good with other researchers results.

The authors pointed out to the study limitation about inclusion of the anteroposterior skeletal relationship influences the chewing-related increase in BBF. 

---Response:  I appreciate your supportive remarks.

I have two thoughts about the research in general:

1- In the introduction and discussion: why didn’t you have any correlation to those syndromes that OPEN is one of the signs and symptoms?!

---Response: Thank you very much for fruitful suggestion. As suggested, we added the information about the relationship between skeletal openbite and syndromes.

2- In the conclusion: How do you know that OPEN has a significant negative impact on BBF?!, What about the possibility that the BBF is the Causative factor of OPEN?!

The references are relatively recent (2021) and relevant in addition to this are well written and arranged and highly related to the study. Is there no references related to the study in the last 3 years?!

---Response: Thank you very much for precious opinion. We just measured brain blood flow in the subjects, and found that the subjects with skeletal openbite exhibited retarded brain blood flow during mastication. That signifies only the positive relationship between skeletal openbite and retarded brain blood flow during mastication, but not cause. Relate to this issue, we previously reported that the mandibular prognathism exhibit retarded brain blood flow during mastication (Sci Rep 2019;9:19104.), similar to that in OPEN observed in the present study. Considering the similar retarded brain blood flow during mastication in both mandibular prognathism and skeletal openbite, we presumed that the retarded brain blood flow would be due to malocclusion, but not causative factor. To prove whether retarded brain blood flow during mastication would be the causative factor of OPEN, extensive additional research is necessary.

As to the references used in the present study, latest paper are not cited. We checked the latest literature again and added some latest papers.

Round 2

Reviewer 1 Report

Comments and Suggestions for Authors

Well done! Good luck from my part.

Reviewer 2 Report

Comments and Suggestions for Authors

Thank you for the corrections. I have still some concerns about sample size